# Associations of age at diagnosis of breast cancer with incident myocardial infarction and heart failure: A prospective cohort study

Jie Liang[1], Yang Pan[1], Wenya Zhang[1], Darui Gao[2,3,4], Yongqian Wang[2,4], Wuxiang Xie[2,3,4]*, Fanfan Zheng[1]*

[1]School of Nursing, Chinese Academy of Medical Sciences & Peking Union Medical College, Beijing, China; [2]Clinical Research Institute, Institute of Advanced Clinical Medicine, Peking University, Beijing, China; [3]Peking University First Hospital, Beijing, China; [4]Key Laboratory of Epidemiology of Major Diseases (Peking University), Ministry of Education, Beijing, China

**\*For correspondence:**
xiewuxiang@hsc.pku.edu.cn
(WX);
zhengfanfan@nursing.pumc.edu.
cn (FZ)

**Competing interest:** The authors declare that no competing interests exist.

## Abstract

**Background:** The associations of age at diagnosis of breast cancer with incident myocardial infarction (MI) and heart failure (HF) remain unexamined. Addressing this problem could promote understanding of the cardiovascular impact of breast cancer.

**Methods:** Data were obtained from the UK Biobank. Information on the diagnosis of breast cancer, MI, and HF was collected at baseline and follow-ups (median = 12.8 years). The propensity score matching method and Cox proportional hazards models were employed.

**Results:** A total of 251,277 female participants (mean age: 56.8 ± 8.0 years), of whom 16,241 had breast cancer, were included. Among breast cancer participants, younger age at diagnosis (per 10-year decrease) was significantly associated with elevated risks of MI (hazard ratio [HR] = 1.36, 95% confidence interval [CI] 1.19–1.56, p<0.001) and HF (HR = 1.31, 95% CI 1.18–1.46, p<0.001). After propensity score matching, breast cancer patients with younger diagnosis age had significantly higher risks of MI and HF than controls without breast cancer.

**Conclusions:** Younger age at diagnosis of breast cancer was associated with higher risks of incident MI and HF, underscoring the necessity to pay additional attention to the cardiovascular health of breast cancer patients diagnosed at younger age to conduct timely interventions to attenuate the subsequent risks of incident cardiovascular diseases.

**Funding:** This study was supported by grants from the National Natural Science Foundation of China (82373665 and 81974490), the Nonprofit Central Research Institute Fund of Chinese Academy of Medical Sciences (2021-RC330-001), and the 2022 China Medical Board-open competition research grant (22-466).

## eLife assessment

In this **valuable** study, the authors sought to investigate the associations of age at breast cancer onset with the incidence of myocardial infarction and heart failure. Based on results from a series of **compelling** statistical analyses, the authors conclude that a younger onset age of breast cancer is associated with myocardial infarction and heart failure, highlighting the need to carefully monitor the cardiovascular status of women who have been diagnosed with breast cancer.

## Introduction

Globally, breast cancer is the most commonly diagnosed cancer in women, with 2.3 million estimated new cases in 2020 (*Sung et al., 2021*). Early screening and treatment advancements have resulted in increasing survival (*American Cancer Society, 2019*; *Berry et al., 2005*) as more than 80% of breast cancer patients survive for at least 5 years worldwide and nearly 50% of patients survive for 10 years in high-income settings (*Allemani et al., 2018*; *UK Cancer Research, 2019*). However, concerns have been raised about long-term comorbidities, particularly cardiovascular disease (CVD), in breast cancer patients, along with the prolonged survival period (*American Cancer Society, 2019*; *Barish et al., 2019*; *Blaes and Konety, 2021*; *Florido et al., 2022*). CVD poses a major threat to health and is the predominant cause of death in breast cancer patients (*Patnaik et al., 2011*), possibly driven by the natural aging process, shared risk factors, and therapy-associated cardiotoxicity (*Khouri et al., 2012*; *Koene et al., 2016*; *Yeh and Bickford, 2009*). For instance, cardiotoxicity induced by anthracycline-based chemotherapy has been repeatedly reported as an adverse effect of breast cancer treatment (*Du et al., 2009*; *Lin and Lengacher, 2019*). Previous studies focused on the subsequent CVD risk after the diagnosis of breast cancer revealed that patients had an excess risk of myocardial infarction (MI) and heart failure (HF) (*Khan et al., 2011*; *Reding et al., 2022*; *Yang et al., 2022*).

Improvements in breast cancer screening also increased the number of patients diagnosed at younger age. The association of breast cancer with HF was found to be affected by age as an increased risk of HF was observed only in younger patients (*Lee et al., 2020*). Thus, we assumed that the age at diagnosis of breast cancer could have an impact on CVD incidence. To date, whether younger age at diagnosis of breast cancer is associated with elevated risks of developing MI and HF remains unexplored. Therefore, by using data from the UK Biobank, of which data on the age at diagnosis of breast cancer and date of incident MI and HF were collected over a relatively long period, this study aimed to investigate the associations of age at diagnosis of breast cancer with subsequent risks of incident MI and HF.

## Methods

### Study design and population

The UK Biobank is a prospective cohort involving sociodemographic and health information of over 270,000 female adults aged 40 and over in the UK. Baseline data were collected between 2006 and 2010. A detailed description of the design and sampling method of the UK Biobank can be found elsewhere (*Hewitt et al., 2016*; *Sudlow et al., 2015*). The UK Biobank has obtained ethical consent from the North West Multi-centre Research Ethics Committee (299116). Written informed consent was received from all participants. This study follows the standards for reporting observational studies (STrengthening the Reporting of OBservational studies in Epidemiology), and a checklist is provided in *Supplementary file 1A*.

*Figure 1* presents the process of participant selection. Briefly, among the 273,325 female adults assessed at baseline, participants with MI or HF at baseline (n = 2992), without complete data on low-density lipoprotein cholesterol (LDL-C; n = 18,912), or having MI or HF before breast cancer during follow-ups (n = 144) were excluded. The remaining 251,277 participants were included in the analysis to investigate the associations of breast cancer with incident MI and HF. Then, 16,241 participants with data on age at diagnosis of breast cancer were included in the analysis to evaluate the associations of age at diagnosis of breast cancer with incident MI and HF. Finally, 16,241 breast cancer participants and their matched healthy controls (1:3; n = 48,723) were included in the propensity score matching analysis to evaluate the associations between breast cancer and incident outcomes among different diagnosis age groups.

### Ascertainment of breast cancer and age at diagnosis of breast cancer

Breast cancer and its diagnosis age were identified using the cancer registry data with the International Classification of Diseases Tenth Revision (ICD-10) codes of C50, which acquire information on cancer diagnoses from a variety of sources, including hospitals, cancer centers, treatment centers, cancer screening programs, hospices, and nursing homes. Detailed information is presented in *Supplementary file 1B*.

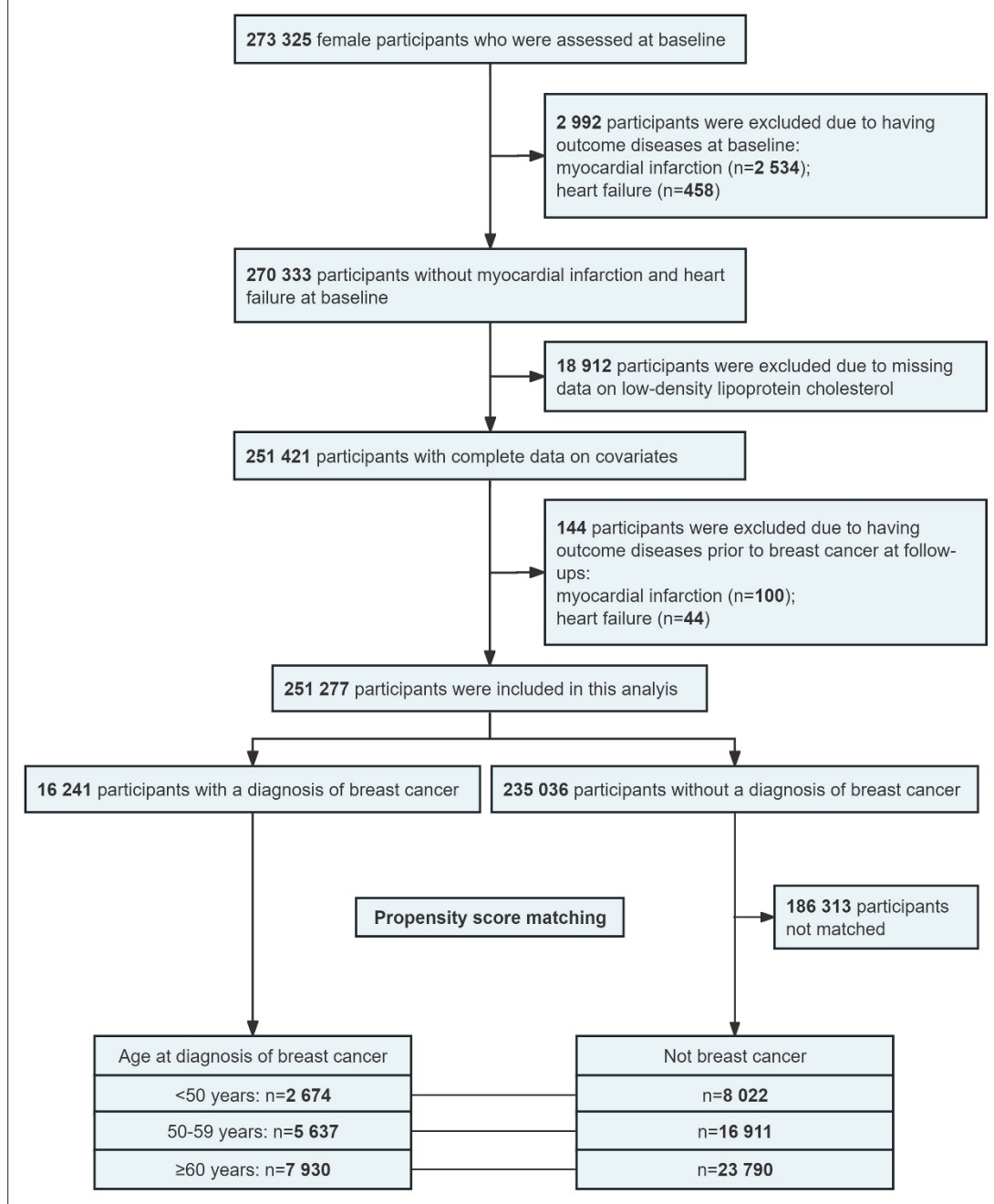

**Figure 1.** Flow chart of participant selection for this study.

The online version of this article includes the following figure supplement(s) for figure 1:

**Figure supplement 1.** Subgroup analyses to identify potential modifying effects from covariates on the associations between breast cancer and incident myocardial infarction in breast cancer participants diagnosed at age <50 and their controls by using Cox proportional hazards models (n = 10,696).

**Figure supplement 2.** Subgroup analyses to identify potential modifying effects from covariates on the associations between breast cancer and incident myocardial infarction in breast cancer participants diagnosed at age 50–59 and their controls by using Cox proportional hazards models (n = 22,548).

**Figure supplement 3.** Subgroup analyses to identify potential modifying effects from covariates on the associations between breast cancer and incident myocardial infarction in breast cancer participants diagnosed at age ≥60 and their controls by using Cox proportional hazards models (n = 31,720).

*Figure 1 continued on next page*

*Figure 1 continued*

**Figure supplement 4.** Subgroup analyses to identify potential modifying effects from covariates on the associations between breast cancer and incident heart failure in breast cancer participants diagnosed at age <50 and their controls by using Cox proportional hazards models (n = 10,696).

**Figure supplement 5.** Subgroup analyses to identify potential modifying effects from covariates on the associations between breast cancer and incident heart failure in breast cancer participants diagnosed at age 50–59 and their controls by using Cox proportional hazards models (n = 22,548).

**Figure supplement 6.** Subgroup analyses to identify potential modifying effects from covariates on the associations between breast cancer and incident heart failure in breast cancer participants diagnosed at age ≥60 and their controls by using Cox proportional hazards models (n = 31,720).

**Figure supplement 7.** Cubic spline curves of the association between diagnosis age of breast cancer and incident myocardial infarction.

**Figure supplement 8.** Cubic spline curves of the association between diagnosis age of breast cancer and incident heart failure.

**Figure supplement 9.** Kaplan–Meier curves of the association between breast cancer and incident myocardial infarction in breast cancer participants diagnosed at age <50 and their controls (n = 10,696).

**Figure supplement 10.** Kaplan–Meier curves of the association between breast cancer and incident myocardial infarction in breast cancer participants diagnosed at age 50–59 and their controls (n = 22,548).

**Figure supplement 11.** Kaplan–Meier curves of the association between breast cancer and incident myocardial infarction in breast cancer participants diagnosed at age ≥60 and their controls (n = 31,720).

**Figure supplement 12.** Kaplan–Meier curves of the association between breast cancer and incident heart failure in breast cancer participants diagnosed at age <50 and their controls (n = 10,696).

**Figure supplement 13.** Kaplan–Meier curves of the association between breast cancer and incident myocardial infarction in participants with heart failure diagnosed at age 50–59 and their controls (n = 22,548).

**Figure supplement 14.** Kaplan–Meier curves of the association between breast cancer and incident heart failure in breast cancer participants diagnosed at age ≥60 and their controls (n = 31,720).

## Ascertainment of MI and HF

Algorithmically defined MI with the date of diagnosis were ascertained using self-reported data, hospital inpatient records, and mortality register data in the UK Biobank. The earliest record of MI from these sources was considered as the date of first occurrence. For HF, the UK Biobank working group has provided the date of first occurrence with the ICD-10 codes of I50. Information on outcomes is detailed in *Supplementary file 1C*. All outcomes were followed up to December 31, 2021.

## Covariates

Covariates included age, ethnicity (white or nonwhite), educational level (higher educational level or not), current smoking (yes or no), current drinking (≥once per week), exercise, obesity, LDL-C, depressed mood, hypertension, diabetes, antihypertensive drug use (yes or no), antidiabetic drug use (yes or no), and statin use (yes or no). A higher educational level was defined as a college or university degree or other professional qualifications. Exercise was defined as participating in moderate or vigorous physical activity for ≥10 min at least twice per week. Depressed mood was defined as a participant reported feeling down, depressed, or hopeless nearly every day or more than half the days over the past 2 weeks. Hypertension was defined as systolic blood pressure (SBP) ≥140 mmHg, diastolic blood pressure (DBP) ≥90 mmHg, self-reported diagnosis of hypertension, or use of antihypertensive medications. Diabetes was defined as glycated hemoglobin ≥ 6.5%, self-reported diagnosis of diabetes, or use of antidiabetic therapy. Obesity was defined as a body mass index ≥ 30 kg/m². Detailed information on covariates is presented in *Supplementary file 1D*.

## Statistical analysis

The analytical baseline used for follow-up was defined as the baseline of UK Biobank (2006–2010). Baseline characteristics are presented as the mean ± standard deviation for continuous variables and frequency (percentage) for categorical variables. The effect sizes of differences in baseline characteristics between participants with and without breast cancer are presented as standardized mean differences for continuous outcomes and the Phi coefficient for dichotomous outcomes, with the standardized mean difference or the Phi coefficient <–0.1 or >0.1 considered significant. Cox proportional hazards models were applied to evaluate the associations of breast cancer and its diagnosis age with incident MI and HF. Years since baseline (2006–2010) to the first occurrence date of MI or HF, death, loss of follow-up, or the end of follow-up (December 31, 2021), whichever came first,

was the time scale in the Cox proportional hazards models. First, we examined the associations of breast cancer with subsequent MI and HF among the total population (n = 251,277). Second, we tested the associations of age at diagnosis of breast cancer with MI and HF among breast cancer participants (n = 16,241). Third, we divided breast cancer participants into three groups according to age at diagnosis:<50 years (n = 2674), 50–59 years (n = 5637), and ≥60 years (n = 7930). Then, three matched healthy controls (1:3) were randomly selected for each breast cancer participant from participants without breast cancer by using the propensity score method (*Parsons, 2004*), which accounted for age, ethnicity, education, current smoking, current drinking, obesity, exercise, LDL-C, depressed mood, hypertension, diabetes, antihypertensive drug use, antidiabetic drug use, and statin use. The associations of breast cancer with incident MI and HF were analyzed in three age groups. The method of matching breast cancer patients with non-breast cancer controls in each diagnosis age group has been used in previous studies (*Cigolle et al., 2022*; *Shang et al., 2021*), as well as in the prior studies of our group (*Liang et al., 2023*; *Wang et al., 2023*; *Zhang et al., 2023*; *Zheng et al., 2024*).

Several sensitivity analyses were performed. First, subgroup analyses were performed to identify potential modifying effects from covariates on the associations of breast cancer and its diagnosis age with incident MI and HF. The Z-test proposed by *Altman and Bland, 2003* was used to compare the difference between the two regression coefficients from subgroup analyses. Second, competing risk models were employed to assess the influence of death as a competing event on the associations of age at diagnosis of breast cancer with incident MI and HF (*Austin and Fine, 2017*). Third, we excluded outcomes that occurred within 5 years since baseline and repeated our main analyses to control for potential reverse causality. Fourth, we restricted analyses to a subgroup of participants aged ≥50 years at baseline since the incidences of MI and HF were relatively low in younger participants. Fifth, we set the deadline for follow-up as December 31, 2019, and repeated main analyses

**Table 1.** Baseline characteristics of the study participants by whether they had a history of breast cancer at baseline or incident breast cancer during follow-up (n = 251,277).

| Characteristic | Breast cancer (n = 16,241) | Non-breast cancer (n = 235,036) | Effect size* |
|---|---|---|---|
| Age (years) | 58.9 ± 7.3 | 56.7 ± 8.0 | 0.275 |
| White | 15,619 (96.2) | 221,292 (94.2) | 0.021 |
| Higher education | 7571 (46.6) | 109,362 (46.5) | <0.001 |
| Current smoking | 1390 (8.6) | 20,764 (8.8) | –0.002 |
| Current drinking | 10,451 (64.4) | 146,679 (62.4) | 0.010 |
| Obesity | 3934 (24.2) | 54,321 (23.1) | 0.007 |
| Exercise | 12,506 (77.0) | 182,852 (77.8) | –0.005 |
| SBP, mmHg | 137.4 ± 19.4 | 135.1 ± 19.2 | 0.118 |
| DBP, mmHg | 81.4 ± 9.9 | 80.7 ± 10.0 | 0.065 |
| HbA$_{1c}$, % | 3.63 ± 0.60 | 3.57 ± 0.59 | 0.093 |
| TC, mmol/L | 5.95 ± 1.14 | 5.88 ± 1.12 | 0.060 |
| HDL-C, mmol/L | 1.60 ± 0.38 | 1.59 ± 0.38 | 0.008 |
| LDL-C, mmol/L | 3.67 ± 0.88 | 3.63 ± 0.87 | 0.049 |
| Depressed mood | 778 (4.8) | 12,351 (5.3) | –0.005 |
| Hypertension | 8643 (53.2) | 112,667 (47.9) | 0.026 |
| Diabetes | 795 (4.9) | 9883 (4.2) | 0.008 |
| Antihypertensive drug use | 3203 (19.7) | 39,767 (16.9) | 0.018 |
| Antidiabetic drug use | 449 (2.8) | 5628 (2.4) | 0.006 |
| Statin use | 1885 (11.6) | 24,502 (10.4) | 0.010 |

The results are presented as the mean ± standard deviation or no. (%).

*The effect sizes are standardized mean differences for continuous outcomes and the Phi coefficient for dichotomous outcomes.

SBP = systolic blood pressure. DBP = diastolic blood pressure. HbA$_{1c}$ = glycated hemoglobin. TC = total cholesterol. HDL-C = high-density lipoprotein cholesterol. LDL-C = low-density lipoprotein cholesterol.

**Table 2.** Associations of breast cancer with incident myocardial infarction and heart failure (n = 251,277).

| Outcome | HR (95% CI) Breast cancer vs. non-breast cancer | p-Value |
|---|---|---|
| **Myocardial infarction** | | |
| Model 1* | 0.84 (0.74–0.95) | 0.005 |
| Model 2† | 0.83 (0.73–0.94) | 0.002 |
| **Heart failure** | | |
| Model 1* | 1.24 (1.12–1.27) | <0.001 |
| Model 2† | 1.20 (1.09–1.33) | <0.001 |

HR, hazard ratio; CI, confidence interval.

*Adjusted for age, ethnicity, and education.

†Further adjusted for current smoking, current drinking, obesity, exercise, low-density lipoprotein cholesterol, depressed mood, hypertension, diabetes, antihypertensive drug use, antidiabetic drug use, and statin use.

to account for the influence of the COVID-19 pandemic on the diagnosis of breast cancer and outcomes since hospital admission and primary care services to chronic diseases were disrupted significantly during this period. Sixth, we further adjusted for menopausal status, breast cancer surgery, and hormone replacement therapy in the main analyses. Among breast cancer participants, 11,460 (70.6%) participants were postmenopausal, 14,255 (87.6%) participants had undergone breast cancer surgery, and 6784 (41.8%) participants had received hormone replacement therapy. Seventh, we added cubic spline curves of the association between the age at diagnosis of breast cancer and incident MI and HF, with the age at diagnosis of breast cancer as a continuous variable. Eighth, we added Kaplan–Meier curves comparing survival probabilities between breast cancer patients and matched controls in each diagnosis age group.

Statistical analyses were performed with SAS 9.4 (SAS Institute, Cary, NC). All analyses were two-sided, with p<0.05 considered significant.

# Results

## Baseline characteristics

A total of 251,277 female participants (mean age: 56.8 ± 8.0 years) were included in the present analyses, of whom 16,241 (6.5%) had breast cancer. The median age at diagnosis was 59 years (interquartile range [IQR]: 52–66 years). *Table 1* shows the baseline characteristics of the participants, grouped by breast cancer status. Overall, breast cancer participants were older and had higher SBP levels.

**Table 3.** Associations of age at diagnosis of breast cancer with incident myocardial infarction and heart failure among breast cancer participants (n = 16,241).

| Outcome | HR (95% CI)* | p-Value |
|---|---|---|
| **Myocardial infarction** | | |
| ≥60 years (n = 7930) | Reference | / |
| 50–59 years (n = 5637) | 1.05 (0.78–1.40) | 0.750 |
| <50 years (n = 2674) | 2.20 (1.54–3.15) | <0.001 |
| Per 10-year decrease | 1.36 (1.19–1.56) | <0.001 |
| **Heart failure** | | |
| ≥60 years (n = 7930) | Reference | / |
| 50–59 years (n = 5637) | 1.32 (1.07–1.64) | 0.010 |
| <50 years (n = 2674) | 1.68 (1.22–2.31) | 0.001 |
| Per 10-year decrease | 1.31 (1.18–1.46) | <0.001 |

*Adjusted for age, ethnicity, education, current smoking, current drinking, obesity, exercise, low-density lipoprotein cholesterol, depressed mood, hypertension, diabetes, antihypertensive drug use, antidiabetic drug use, and statin use.

HR = hazard ratio. CI = confidence interval.

**Table 4.** Associations of breast cancer with incident myocardial infarction and heart failure among different diagnosis age groups after propensity score matching (n = 64,964).

| Outcome | HR (95% CI)*<br>Breast cancer vs. non-breast cancer | p-Value |
| --- | --- | --- |
| **Myocardial infarction** | | |
| ≥60 years (n = 31,720) | 0.75 (0.63–0.89) | 0.001 |
| 50–59 years (n = 22,548) | 0.75 (0.58–0.97) | 0.028 |
| <50 years (n = 10,696) | 1.75 (1.21–2.52) | 0.003 |
| **Heart failure** | | |
| ≥60 years (n = 31,720) | 1.03 (0.90–1.19) | 0.650 |
| 50–59 years (n = 22,548) | 1.38 (1.13–1.69) | 0.002 |
| <50 years (n = 10,696) | 2.21 (1.55–3.17) | <0.001 |

*Adjusted for age, ethnicity, education, current smoking, current drinking, obesity, exercise, low-density lipoprotein cholesterol, depressed mood, hypertension, diabetes, antihypertensive drug use, antidiabetic drug use, and statin use.
HR = hazard ratio. CI = confidence interval.

## Associations of breast cancer with incident MI and HF

During a median follow-up of 12.8 years (IQR 12.1–13.6 years), 4549 (1.8%) MI and 4917 (2.0%) HF were identified. As presented in *Table 2*, after adjusting for multiple covariates among the total population (n = 251,277), breast cancer participants exhibited a significantly lower risk of developing MI (hazard ratio [HR] = 0.83, 95% confidence interval [CI] 0.73–0.94, p=0.002) and a significantly higher risk of developing HF (HR = 1.20, 95% CI 1.09–1.33, p<0.001).

## Associations of age at diagnosis of breast cancer with incident MI and HF among breast cancer participants

As shown in *Table 3*, among 16,241 breast cancer participants, age at diagnosis was significantly associated with subsequent risks of MI and HF; that is, those diagnosed at younger ages had higher risks of developing MI (per 10-year decrease: HR = 1.36, 95% CI 1.19–1.56, p<0.001) and HF (per 10-year decrease: HR = 1.31, 95% CI 1.18–1.46, p<0.001).

## Associations of breast cancer with incident MI and HF among different diagnosis age groups based on propensity score matching data

We then further investigated the relationships between breast cancer and incident outcomes in different diagnosis age groups among 16,241 breast cancer participants and their matched controls without breast cancer by using propensity score matching analyses. As presented in *Supplementary file 1E*, after propensity score matching, no significant difference was detected between participants with and without breast cancer in all covariates. *Table 4* shows that breast cancer diagnosed before 50 years was associated with the highest HRs for incident MI and HF compared with those without breast cancer (MI: HR = 1.75, 95% CI 1.21–2.52, p=0.003; HF: HR = 2.21, 95% CI 1.55–3.17, p<0.001), followed by breast cancer diagnosed between 50 and 59 years (MI: HR = 0.75, 95% CI 0.58–0.97, p=0.028; HF: HR = 1.38, 95% CI 1.13–1.69, p=0.002), and then breast cancer diagnosed at 60 years and over (MI: HR = 0.75, 95% CI 0.63–0.89, p=0.001; HF: HR = 1.03, 95% CI 0.90–1.19, p=0.650).

### Sensitivity analysis

As shown in *Figure 1—figure supplements 1–6*, the results from subgroup analyses were similar to those from our main analyses. Interestingly, our subgroup analyses found that hypertension and diabetes modified the association of breast cancer with incident HF. Cubic spline curves of the association between age at diagnosis of breast cancer and incident MI and HF with age at diagnosis of breast cancer as a continuous variable and Kaplan–Meier curves comparing survival probabilities between breast cancer patients and matched controls in each diagnosis group were consistent with our main results (*Figure 1—figure supplements 7–14*). As presented in *Supplementary file 1F–O*, the results

remained stable after further adjusting for competing risk of death, excluding participants diagnosed with MI or HF within 5 years since baseline, excluding participants aged <50 years at baseline, setting the deadline of follow-up as December 31, 2019, or further adjusting for menopausal status, breast cancer, and hormone replacement therapy.

## Discussion

By using data from the large, prospective cohort of the UK Biobank, this study revealed that breast cancer patients were at a decreased risk of developing MI and an increased risk of developing HF compared to participants without breast cancer. It is worth noting that younger age at diagnosis of breast cancer was associated with elevated risks of incident MI and HF among breast cancer patients. After propensity score matching, the strength of the associations gradually increased with descending age at diagnosis of breast cancer.

Our findings were in line with several recent studies investigating the associations of breast cancer with CVD (*Gue et al., 2022*; *Strongman et al., 2019*). A large cohort study linked to the UK Clinical Practice Research Datalink revealed a decreased risk of incident MI and an increased risk of incident HF in breast cancer patients during an average 7-year follow-up, with an HR of 0.82 and 1.13, respectively, which was close to our results (*Strongman et al., 2019*). Another population-based study at the nationwide level in France, with a mean follow-up of 5 years, also demonstrated similar risks of MI and HF (*Gue et al., 2022*). The elevated risk of subsequent HF in breast cancer patients has been observed in most studies, which may be related to the cardiotoxic effect on the myocardium induced by anthracycline and trastuzumab (*Abdel-Qadir et al., 2019*; *Gue et al., 2022*; *Lee et al., 2020*; *Reding et al., 2022*; *Strongman et al., 2019*; *Yang et al., 2022*). The decreased risk of MI observed in the study might be driven by the cardioprotective effects of tamoxifen and socioeconomic factors, as indicated in previous studies (*Grainger and Schofield, 2005*; *Khosrow-Khavar et al., 2017*; *Tweed et al., 2018*), while the precise mechanisms have not yet been elucidated. However, previous findings concerning MI or ischemic heart disease were inconsistent, with some researchers reporting increased risk (*American Cancer Society, 2019*; *Rugbjerg et al., 2014*), but others observing null results (*Abdel-Qadir et al., 2019*; *Khan et al., 2011*). The discordance may be partly due to heterogeneity in population characteristics, length of follow-ups, and sources of breast cancer and CVD data. For example, the age of the population might influence the association, as indicated by a nationwide cohort study in Korea, which showed that an increased risk of HF only existed in younger breast cancer patients (<50 years) (*Lee et al., 2020*). Moreover, the present study extended prior research by comparing the risk of MI and HF among different diagnosis age groups and demonstrated that age at diagnosis of breast cancer was an important determinant for the risks of incident MI and HF.

To the best of our knowledge, this is the largest study to date to explore the impact of the age at diagnosis of breast cancer on the subsequent risks of MI and HF. With accurate data on age at diagnosis of breast cancer and sufficient CVD events over the lengthy follow-up, this study revealed that risks of incident MI and HF increased with decreasing age at diagnosis of breast cancer (especially <50 years). A Danish cohort study focused on patients of adolescent and young adult cancer indicated that breast cancer patients diagnosed at younger ages, particularly between 20 and 24 years, had the highest risk of CVD (*Rugbjerg et al., 2014*). Similarly, the Teenage and Young Adult Cancer Survivor Study demonstrated that cardiac death risk increased in younger diagnosis age groups, although the trend was not significant (*Henson et al., 2016*). Possibly due to a relatively small sample size, a narrow range of diagnosis age, and insufficient CVD events, the trend between age at diagnosis of breast cancer and CVD risk was not detected in the two studies, and the results need further verification. The present study addressed these issues by focusing on a population with an age at diagnosis ranging from 27 to 82 years, while the prior studies were both limited to 15–39 years and with nearly 10,000 participants diagnosed with MI or HF during the follow-up, providing a robust and reliable finding on the associations of the age at diagnosis of breast cancer with MI and HF.

Although the mechanisms underlying the associations are still not fully understood, several potential pathways have been proposed. First, younger age was found to be associated with a higher tumor grade and a more aggressive phenotype (e.g., triple-negative breast cancer) in breast cancer patients (*Colleoni et al., 2002*; *El Saghir et al., 2006*; *McGuire et al., 2015*), for whom anthracycline-containing chemotherapy was the usual therapeutic regimen. Anthracycline is known to confer great cardiotoxicity by harming cardiac myocytes at a cumulative dose, resulting

in subsequent CVD (*Greenlee et al., 2022*; *Hooning et al., 2007*; *Smith et al., 2010*). Second, patients diagnosed at younger ages tended to choose breast-conserving surgery, often accompanied by radiotherapy, which was associated with an accelerated coronary calcium burden (*Lai et al., 2021*). A dose–response relationship has been observed between radiotherapy and CVD, with major coronary events increasing by 7.4% per gray (*Darby et al., 2013*). Third, younger premenopausal patients were at risk of early menopause induced by chemotherapy (*Zavos and Valachis, 2016*), leading to reduced exposure to the cardioprotective effects of estrogen and an elevated risk of developing CVD (*Mendelsohn and Karas, 1999*; *Wellons et al., 2012*; *Xu et al., 2006*). Furthermore, hormone replacement therapy in postmenopausal women was also associated with increased CVD risk (*Rossouw et al., 2002*).

This study has several strengths. First, data on the diagnosis of MI and HF were from hospital inpatient records, mortality register data, primary care data, and self-reported data (*UK Biobank Follow-up and Outcomes Working Group, 2022*). These multisource data outweighed self-reported data with higher accuracy as approximately 30% of self-reported CVD cases were misclassified by patients (*Barr et al., 2009*). Second, the large sample size of the UK Biobank ensured a robust conclusion on the associations of the age at diagnosis of breast cancer with MI and HF with sufficient statistical power. Third, propensity score matching analyses after controlling for multiple traditional risk factors significantly reduced confounding bias.

Despite these strengths, the present study has some limitations. First, a causal relationship cannot be concluded due to the observational study design. Second, 22,048 participants were excluded, which might lead to selection bias. Significant differences in LDL-C levels existed between participants included and excluded (*Supplementary file 1P*). In general, the participants included were younger and healthier, which might bias the associations observed in this study. Third, even though we have adjusted for many traditional confounders, the possibility of residual confounding cannot be ruled out. For instance, due to a lack of information on the staging of breast cancer, radiotherapy, and chemotherapy of breast cancer patients in the UK Biobank, we were unable to assess the impact of these potential confounders. Fourth, the study population primarily consisted of the white ethnicity with a proportion of 94.3%, which may not represent the general UK population. Thus, generalization of the findings should be cautious, and validations in other populations are needed.

## Conclusion

This study demonstrated that younger age at diagnosis of breast cancer was associated with higher risks of incident MI and HF. This finding has significant implications for clinical and public health, highlighting the need to pay attention to the cardiovascular health of younger breast cancer patients, especially those diagnosed before 50 years. Further studies concerning this aspect are warranted and cardiac monitoring strategies need to be developed based on the cardiovascular risk of these individuals.

## Acknowledgements

We appreciate the efforts made by the original data creators, depositors, copyright holders, the funders of the data collections, and their contributions to access to data from the UK Biobank, approved project number 90492.

## Additional information

### Funding

| Funder | Grant reference number | Author |
| --- | --- | --- |
| National Natural Science Foundation of China | 82373665 | Fanfan Zheng |
| National Natural Science Foundation of China | 81974490 | Wuxiang Xie |

| Funder | Grant reference number | Author |
|---|---|---|
| Nonprofit Central Research Institute Fund of Chinese Academy of Medical Sciences | 2021-RC330-001 | Fanfan Zheng |
| 2022 China Medical Board-open competition research grant | 22-466 | Fanfan Zheng |

The funders had no role in study design, data collection and interpretation, or the decision to submit the work for publication.

### Author contributions
Jie Liang, Conceptualization, Data curation, Formal analysis, Investigation, Methodology, Writing - original draft, Writing – review and editing; Yang Pan, Wenya Zhang, Darui Gao, Yongqian Wang, Writing – review and editing; Wuxiang Xie, Fanfan Zheng, Conceptualization, Supervision, Funding acquisition, Investigation, Methodology, Writing – review and editing

### Author ORCIDs
Jie Liang ⓘ https://orcid.org/0000-0002-3613-8488
Fanfan Zheng ⓘ https://orcid.org/0000-0003-2767-2600

### Ethics
The UK Biobank has received ethical consent from the North West Multi-centre Research Ethics Committee (MREC) (299116). Written informed consent was obtained from all participants. This research was done without participants' involvement.

**Reviewer #1 (Public Review):** https://doi.org/10.7554/eLife.95901.3.sa1
Author response https://doi.org/10.7554/eLife.95901.3.sa2

---

# Additional files

### Supplementary files
• Supplementary file 1. STROBE Statement—Checklist, ascertainment of variables, comparison of baseline characteristics by breast cancer status after propensity score matching, sensitivity analysis, and comparison of baseline characteristics between participants included and excluded. (**A**) STROBE Statement—Checklist of items that should be included in reports of cohort studies. (**B**) Ascertainment of breast cancer and age at breast cancer diagnosis. (**C**) Ascertainment of myocardial infarction and heart failure. (**D**) Definition and assessment of covariates. (**E**) Baseline characteristics of participants by breast cancer status after propensity score matching (n = 64,964). (**F**) Associations of age at breast cancer diagnosis with incident myocardial infarction and heart failure among breast cancer participants: competing risk models (n = 16,241). (**G**) Associations of breast cancer with incident myocardial infarction and heart failure among different diagnosis age groups after propensity score matching: competing risk models (n = 64,964). (**H**) Associations of age at breast cancer diagnosis with incident myocardial infarction and heart failure among breast cancer participants after excluding myocardial infarction and heart failure diagnosed within 5 years since baseline (n = 15,589). (**I**) Associations of breast cancer with incident myocardial infarction and heart failure among different diagnosis age groups after excluding myocardial infarction and heart failure diagnosed within 5 years since baseline, results from propensity score matching analyses (n = 62,356). (**J**) Associations of age at breast cancer diagnosis with incident myocardial infarction and heart failure among breast cancer participants after excluding participants aged <50 years at baseline (n = 14,000). (**K**) Associations of breast cancer with incident myocardial infarction and heart failure among different diagnosis age groups after excluding participants aged <50 years at baseline, results from propensity score matching analyses (n = 56,000). (**L**) Associations of age at breast cancer diagnosis with incident myocardial infarction and heart failure among breast cancer participants when the follow-up period ends on December 31, 2019 (n = 15,909). (**M**) Associations of breast cancer with incident myocardial infarction and heart failure among different diagnosis age groups when the follow-up period ends on December 31, 2019, results from propensity score matching analyses (n = 63,636). (**N**) Associations of age at breast cancer diagnosis with incident

myocardial infarction and heart failure among breast cancer participants after further adjusting for menopausal status, breast cancer surgery, and hormone replacement therapy (n = 16,241). (**O**) Associations of breast cancer with incident myocardial infarction and heart failure among different diagnosis age groups after further adjusting for menopausal status, breast cancer surgery, and hormone replacement therapy, results from propensity score matching analyses (n = 64,964). (**P**) Comparison of baseline characteristics between participants included (n = 251,277) and excluded due to history of myocardial infarction or heart failure, without complete data on low-density lipoprotein cholesterol, or having myocardial infarction or heart failure before breast cancer at follow-ups (n = 22,048).

- MDAR checklist

## Data availability

The data used for analysis in this study is available from UK Biobank project site, subject to registration and application process. Further details can be found at https://www.ukbiobank.ac.uk.

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
