## [Editor Report · eLife assessment]

In this **valuable** study, the authors sought to investigate the associations of age at breast cancer onset with the incidence of myocardial infarction and heart failure. Based on results from a series of **compelling** statistical analyses, the authors conclude that a younger onset age of breast cancer is associated with myocardial infarction and heart failure, highlighting the need to carefully monitor the cardiovascular status of women who have been diagnosed with breast cancer.

---

## [Referee Report · Reviewer #1 (Public Review)]

Summary:

The authors sought to investigate the associations of age at breast cancer onset with the incidence of myocardial infarction (MI) and heart failure (HF). They employed a secondary data analysis of the UK Biobank. They used descriptive and inferential analysis including Cox proportional hazards models to investigate the associations. Propensity score matching was also used. They found that Among participants with breast cancer, younger onset age was significantly associated with elevated risks of MI (HR=1.36, 95%CI: 1.19 to 1.56, P<0.001) and HF (HR=1.31, 95% CI: 1.18 to 1.46, P<0.001). the reported similar findings after propensity matching.

Strengths:

The use of a large dataset is a strength of the study as the study is well-powered to detect differences. Reporting both the unmatched and the propensity-matched estimates was also important for statistical inference.

Weaknesses:

The authors have addressed all my previous comments. I have no further comments.

---

## [Author Response]

The following is the authors’ response to the original reviews.

**Response to Reviewer #1:**
Comment 1:Summary:The authors sought to investigate the associations of age at breast cancer onset with the incidence of myocardial infarction (MI) and heart failure (HF). They employed a secondary data analysis of the UK Biobank. They used descriptive and inferential analysis including Cox proportional hazards models to investigate the associations. Propensity score matching was also used. They found that Among participants with breast cancer, younger onset age was significantly associated with elevated risks of MI (HR=1.36, 95%CI: 1.19 to 1.56, *P*<0.001) and HF (HR=1.31, 95% CI: 1.18 to 1.46, *P*<0.001). the reported similar findings after propensity matching.Strengths:The use of a large dataset is a strength of the study as the study is well-powered to detect differences. Reporting both the unmatched and the propensity-matched estimates was also important for statistical inference.Weaknesses:Despite the merits of the paper, readers may get confused as to whether authors are referring to “age at breast cancer onset” or “age at breast cancer diagnosis”. I suppose the title refers to the latter, in which case it will be best to be consistent in using “age at breast cancer diagnosis” throughout the manuscripts. I would recommend a revision to the title to make it explicit that the authors are referring to “age at breast cancer diagnosis”.

Thank you for your nice comments and suggestions. Yes, as you mentioned, in this study, we focused on age at breast cancer diagnosis, which was obtained from the cancer registry data in the UK Biobank and was used in all the analyses. We agree with you that it would be better to consistently use “age at diagnosis of breast cancer” throughout the manuscripts for a better understanding; therefore, we have replaced “age at breast cancer onset” with “age at diagnosis of breast cancer”.

Change in the manuscript:

“Age at breast cancer onset” was replaced with “age at diagnosis of breast cancer” in the title and throughout the manuscripts.

**Recommendations For The Authors:**
Kindly review the references for the location of the full stop. Putting the full stop at the end of the parenthesis makes reading smother than its current form as it is difficult to know when the new sentence begins.

Thank you for your suggestion. We have made revisions to the location of the full stop next to a reference.

Change in the manuscript:

The full stop was put at the end of the parenthesis of a reference throughout the manuscripts.

**Response to Reviewer #2:**
Comment 1:This is a well-presented large analysis from the UK Biobank of nearly 250,000 female adults. The authors examined the associations of breast cancer diagnosis with incident myocardial infarction and heart failure by different onset age groups. Based on results from a series of statistical analyses, the authors concluded that younger onset age of breast cancer was associated with myocardial infarction and heart failure, highlighting the necessity of careful monitoring of cardiovascular status in women diagnosed with breast cancer, especially those younger ones.Comments to consider:It’s thoughtful for the authors to have included and adjusted for menopausal status, breast cancer surgery, and hormone replacement therapy in their sensitivity analysis. It would be informative if the authors presented the number and percentages of menopause and cancer treatments.

Thank you for your comments. As suggested, we have provided more detailed information on the number and percentage of menopausal status and breast cancer treatments.

Change in the manuscript:

Page 11, Lines 208 to 211: added “Among participants with breast cancer, 11 460 (70.6%) participants were postmenopausal, 14 255 (87.6%) participants had undergone breast cancer surgery, and 6 784 (41.8%) participants had received hormone replacement therapy.”

Change in the supplementary material:

The number and percentage of menopausal status, breast cancer surgery, and hormone replacement therapy were added to Table S13.

aAdjusted for age, ethnicity, education, current smoking, current drinking, obesity, exercise, low-density lipoprotein cholesterol, depressed mood, hypertension, diabetes, antihypertensive drug use, antidiabetic drug use, statin use, menopausal status, breast cancer surgery, and hormone replacement therapy.

HR, hazard ratio; CI, confidence interval.

Comment 2:The analytical baseline used for follow-up should be pointed out in the methods section. It’s confusing whether the analytic baseline was defined as the study baseline or the time at breast cancer diagnosis.

We apologize for the confusion. In this study, the analytical baseline used for follow-up was defined as the baseline of UK Biobank (2006-2010) and we have pointed it out in the methods section as suggested.

Change in the manuscript:

Page 9, Lines 165 to 166: added: “The analytical baseline used for follow-up was defined as the baseline of UK Biobank (2006-2010).”

Comment 3:Did the older onset age group have a longer follow-up duration? Could the authors provide information on the length of follow-up by age of onset in Supplementary Table S4? It would give the readers more information regarding different age groups.

Thank you for your question. We compared the time of follow-up among the three diagnosis age groups and found that although the durations of follow-up among the three groups were quite similar (as shown in Table S4), statistical analysis revealed a significant difference with the older diagnosis age group demonstrating a longer follow-up duration (*P* for Kruskal-Wallis test <0.001). This is understandable as with large sample sizes, even a slight difference could lead to statistical significance. According to your suggestion, we have added information on the length of follow-up by age of diagnosis in Supplementary Table S4.

Change in the supplementary material:

Added the median and interquartile range of follow-up in Supplementary Table S4.

The results are presented as the mean ± standard deviation, or No. (%).

aThe effect sizes are standardized mean differences for continuous outcomes and the Phi coefficient for dichotomous outcomes.

LDL-C, low-density lipoprotein cholesterol.